# Utilization of a Diet Optimization Model in Ensuring Adequate Intake among Pregnant Women in Selangor, Malaysia

**DOI:** 10.3390/ijerph16234720

**Published:** 2019-11-27

**Authors:** Zeenat Begam Sawal Hamid, Roslee Rajikan, Siti Masitah Elias, Nor Aini Jamil

**Affiliations:** 1Dietetic Programme, Faculty of Health Sciences, Universiti Kebangsaan Malaysia, Kuala Lumpur 50300, Malaysia; drzeenatbegam@gmail.com (Z.B.S.H.); ainijamil@ukm.edu.my (N.A.J.); 2Financial Mathematics Programme, Faculty of Science and Technology, Universiti Sains Islam Malaysia, Nilai 71800, Negeri Sembilan, Malaysia; masitah@usim.edu.my

**Keywords:** pregnant women, linear programming, diet optimization model, healthy diet

## Abstract

Achieving nutritional requirements for pregnant women in rural or suburban households while maintaining the intake of local and culture-specific foods can be difficult. Usage of a linear programming approach can effectively generate diet optimization models that incorporate local and culturally acceptable menus. This study aimed to determine whether a realistic and affordable diet that achieves recommended nutrient intakes for pregnant women could be formulated from locally available foods in Malaysia. A cross-sectional study was conducted to assess the dietary intake of 78 pregnant women using a 24-h dietary recall and a 3-day food record. A market survey was also carried out to estimate the cost of raw foods that are frequently consumed. All linear programming analyses were done using Excel Solver to generate optimal dietary patterns. Our findings showed that the menus designed from diet optimization models using locally available foods would improve dietary adequacy for the seven food groups based on the Malaysian Dietary Guidelines 2010 (MDG 2010) and the 14 nutrients based on Recommended Nutrient Intake 2017 (RNI 2017) in pregnant women. However, inadequacies remained for iron and niacin, indicating that these nutrients may require supplementation.

## 1. Introduction

Maternal nutrition is critical for both maternal and child health in a period known as the first 1000 days, from the time of conception until two-years post-partum [1]. One of the important risk factors for maternal mortality and foetal growth restriction is undernutrition during pregnancy. It increases the risk of neonatal deaths and contributes to impaired post-natal linear growth and development [1,2]. Evidence from research also shows that maternal malnutrition has long-lasting programming effects on the risk of developing non-communicable diseases later in life [3].

As for the women, adequate intake of nutrients during pregnancy reduces the risk of maternal morbidity including anaemia, unhealthy gestational weight gain, gestational diabetes mellitus (GDM), preeclampsia, preterm births, and miscarriages [4]. Among pregnant women in Malaysia, 80–90% were reported to have iron deficiency and 38–42% developed anaemia [5]. The prevalence of GDM in Malaysia ranges between 18.3% and 27.9% [6] and the number is expected to increase. Nutrition therapy remains an important approach in managing GDM [7].

A study published in 2016, revealed that rice, sugar and sweetened condensed milk were among the top food items consumed daily by Malaysian adults [8]. Pregnant women in Malaysia had inadequate intake of fruits and vegetables compared to the Malaysian Dietary Guidelines (MDG) [9], similar to the intake pattern observed among Polish pregnant women [10]. Most vegetables and fruits were avoided due to food taboos and lack of nutrition knowledge [11,12].

To ensure sufficient nutrient intake among Malaysian pregnant women, it is best to follow recommendations by the MDG and Recommended Nutrient Intake (RNI) [13,14,15]. The MDG provides guidelines to choose food according to food groups whilst RNI emphasizes on specific macro and micronutrients as well as calories for each trimester of pregnancy.

Diet optimization model based on linear programming has been found to provide solution for the complex problem in designing diets that meet nutrient recommendations while maintaining the local food habits and the intake of culture-specific foods [16]. Several studies have successfully constructed local food recommendations for pregnant women, which achieved sufficient nutrient intake using linear programming. These culture specific studies are from China [17], Nigeria [18], Burkina Faso in West Africa [19], and Tanzania [20].

Until now, there has been no research available suggesting a palatable menu using local foods to aid Malaysian pregnant women to achieve optimal nutrient intake. Hence, the aim of this study was to explore the possibility of developing a palatable and culture specific diet, using a diet optimization model, which achieves the national recommendations with minimal cost for pregnant women in the selected area.

## 2. Materials and Methods

### 2.1. Study Design and Participants

This cross-sectional study used convenience sampling at a suburban clinic for maternal and child health care services in Selangor, Malaysia, between August 2018 and March 2019. This study involved 78 pregnant women aged between 20 and 45 years. The inclusion criteria were pregnant women with >6 weeks of gestation period and parity less than 7. Those with health conditions requiring specific diet recommendation such as GDM were excluded. This study gained approval from the Research Ethics Committee of University Kebangsaan Malaysia (UKM) (FSK NN-2019-119). All participants gave their written informed consent prior to participating in this study.

### 2.2. Data Collection

For dietary intake a 24-h dietary recall and a three-day food record of the participants were obtained to assess eating patterns. The 24-h diet recall was done through interviews by the researcher, while the three-day food record was performed by the participants themselves by documenting their food intake for three days in a week including either Saturday or Sunday. Records of the diet intake were collected from the subjects to evaluate the percentage of subjects who achieved the RNI.

Nutritionist pro software was used to determine the nutrient contents of the food items, using the Malaysian food databases [21]. There were 783 food items analysed for energy and macronutrients (protein, fat, and carbohydrate), minerals (calcium, phosphorus, iron, sodium, and potassium), vitamins (vitamin A, niacin, riboflavin, thiamine, and vitamin C) and fibre. The information was fed into the diet optimization model and was used as a comparison to the nutrient intake of the subjects. Recommended Nutrient Intake 2017 [15] and Malaysian Dietary Guidelines 2010 [14] were used to determine the nutrient requirement of pregnant women according to each trimester.

### 2.3. Market Survey

Market surveys were conducted at nearby local grocery shops, wet markets (markets selling fresh meat, fish, produce and other perishable goods) and supermarkets. The food items were based on the food recorded by the participants. The food prices were based on price per serving size.

### 2.4. Data Analysis

All analyses were conducted using the Statistical Package for Social Sciences (SPSS) version 20.0 (IBM, Armonk, NY, USA). For categorical data, the results were presented descriptively and for continuous data, means and standard deviation were reported if they were normally distributed. The diet optimization model constructed using linear programming was developed by utilizing Microsoft Excel with OpenSolver plug in developed by Mason [22]. It aims to suggest the number of servings for each food item that satisfies the recommended nutrient intakes with the lowest cost [23].

### 2.5. Diet Optimization Model

According to previous study [24], the mathematical formula of the model is as below:Min *p* = *∑ c_j_ x_j_*(1)
subject to:*l_i_ ≤ ∑ a_ij_ x_j_ ≤ u_i_*(2)
*x_j_ ≥ 0, x_j_ ∈ Z*

The objective of the model was to minimize food cost, *p*. The portion size of food item *j* is represented as *x_j_*; *a_ij_* denotes the amount of nutrient *i* in one portion of food item *j*; *c_j_* is the cost of one portion of food item *j*; *l_i_* and *u_i_* denote the smallest and largest acceptable quantity of nutrient *i* respectively; and the last constraint specifies the portion size, *x_j_*, to be an integer value.

The MDG 2010 and RNI 2017 were the constraints in the model for this study. The lower bound and upper bound limits of all the nutrients were set based on RNI 2017. Besides, the upper limit and lower limit of the portion size for each food item and food group were also adjusted according to MDG 2010 and the common intake pattern of the participants to ensure that the suggested menus were suitable for them. The food items were grouped into seven food groups as in MDG 2010, which are cereals and grains, fruits, vegetables, meat/poultry, fish, legumes, and milk and dairy products. The MDG recommends pregnant women to follow the food pyramid for adults. To ensure the developed menus were palatable, oil, sugar and salt were added into the diet.

These constraints, together with food prices and nutrient composition were fed into the diet optimization model. Once the software ran, the selected food items with its cost were considered as a model. The selected food items were then constructed into palatable menus.

## 3. Results and Discussion

### 3.1. Socio-Demographic Data

A total of 78 participants (mean age: 31.2 years old, range: 20–45 years) took part in this study (Table 1). The majority of the participants were Malays (98.7%), aged between 30 and 39 years old (58.8%) with a tertiary education (56.4%). About one-third of the participants had monthly income ranging US$262.78–716.67 (33.3%) and US$740.56–1194.6 (35.9%), while the remaining 25.6% had household income above US$1194.46 (RM 1.00 = USD 0.24). The majority of the participants (52.3%) were at trimester 2 (13–26 weeks of gestation) and had normal pre-pregnancy Body Mass Index (BMI)(41%).

### 3.2. Haemoglobin Level and Supplement Consumption

In this study, 29.5% of the participants were anaemic with haemoglobin (Hb) level less than 11 g/dL. Out of these, 7.7% had Hb level less than 10 g/dL, which is categorized as moderate anaemia. This percentage is slightly less compared to a finding in a previous study done in 2015 which found that 38–42% of pregnant women in Malaysia developed anaemia [5]. In this study, 85.9% of the participants consumed iron and/or vitamin supplements during current pregnancy. Ninety-four percent of the participants with normal haemoglobin level consumed an iron supplement. As for the anaemic group, 79% did not consume iron supplement. Supplement consumption during pregnancy was positively associated with Hb levels (*p* < 0.05). Another study also found that iron supplementation was statistically associated with Hb levels among pregnant women [25].

### 3.3. Dietary Intake

The calories and nutrient intake with the percentage of subjects achieving RNI is shown in Table 2. The mean intake of energy for participants was 2310 kcal. Only 48.7% of the subjects met the RNI requirements for energy intake. This is similar to a study conducted among Jordanian mothers [26]. However, it is higher as compared to results reported by Manaf et al. where only 21.9% of the respondents achieved the RNI requirement for energy [27]. Practices of food beliefs and food taboos [11,27] especially among Malay pregnant women, may lead to decreased food intake which may cause lower maternal weight gain [28].

The mean percentages of energy contributed from carbohydrate, protein and fat was 53.6%, 14.9%, and 30.1%, respectively. Less than half (40%) of the subjects met the nutrient requirement for fat intake while the majority met the recommendations for carbohydrates (96%) and protein (87%). Cheng et al. saw similar results in their study where a higher percentage of study population consumed more carbohydrates and less fat compared to the recommended nutrient intake [29]. The low intake of fat among participants in the current study could be due to increased nutrition awareness as the pregnant women have been exposed to dietary counselling during their routine antenatal check-up. However, a meta-analysis of numerous cohort studies from developed countries found that the total fat and saturated fat consumption were higher than the recommendation [30].

For vitamin intake, majority of the subjects (80.7%) reached the requirement for vitamin A, followed by thiamine (65.3%), vitamin C (64.1%) and riboflavin (41%). However, only 11.5% achieved the requirement for niacin. Manaf et al. reported similar findings for vitamins B1, B2, and C [27]. Looking at minerals, the highest percentage of subjects achieving RNI requirement is 98.7% for sodium, and this was followed by phosphorus (97.4%). For iron, only two-thirds of the participants met RNI requirement (70.5%). Calcium and potassium both had a very low percentage of subjects achieving RNI, with 2.6% and 1.3%, respectively. Another study showed similar data for calcium and fibre intake [31].

### 3.4. Development of Healthy and Balanced Menus Which Achieves MDG 2010 and RNI 2017

The development of healthy and balanced menus that ensure sufficient nutrient intake for Malaysian pregnant women involves the fulfilment of RNI 2017 [15] and MDG 2010. Table 3 shows the comparison of food groups by the three Diet Optimization Models (one model for each trimester) with MDG and palatability. These models are based on RNI 2017, MDG 2010 and palatability as the constraints for each trimester. These models should satisfy the upper and lower limits of the constraints. The comparison of the models with RNI 2017 is shown in Table 4. For MDG 2010, the constraints include the upper and lower limits of the serving size of seven types of food groups, which are cereals and grains, fruits, vegetables, meat and poultry, fish, legumes, milk and dairy products.

Palatability constraints were inserted in the model to ensure that certain food groups do not exceed the food item normally consumed and generally accepted by the participants as the linear programming model finds the combinations of the cheapest food items that fulfil all nutrient requirements. Palatability in this study is defined as foods that were usually consumed by the participants, which were locally available and chosen due to personal preference. In this study, the foods selection reported in the 24-h dietary recall and 3-day food records were hypothesized to be due to personal preferences.

Constraints were also added for the number of each food type allowed; such as only one serving of ‘patin’ (silver catfish) fish was allowed for the one-day menu to avoid repetition of the same food in the meals which would also affect palatability, as repetition of the same food in the meals in a day would cause boredom.

The Diet Optimization Model in this study has successfully generated models where the constraint values are all within the range of MDG 2010, palatability, and RNI 2017. Sodium and vitamin A reached the upper limit of the maximum acceptable value of RNI for these constraints. Energy, protein, carbohydrate, fat, vitamins B1, B2, and C as well as phosphorus were in the moderately acceptable value of RNI constraint. Meanwhile, the other nutrients (fibre, potassium, calcium, iron and vitamin B3) reached only the lower limit of RNI.

In the present study, the menus were constructed using raw food materials with the lowest cost. The menus were developed using four diet optimization models (three without supplements, one with supplements) for each trimester. The menus are shown in Table 5, Table 6 and Table 7. It was estimated that the minimal food cost for pregnant women without supplements for the first trimester is US$1.58 and with free supplements is US$1.56. For trimester 2, the cost is US$1.61, and with free supplements US$1.47. The price for a one-day menu for pregnant women at the second trimester including free supplements is less than the first trimester because only folic acid is provided as a supplement in the first trimester while the pregnant women are given folic acid, iron, a vitamin B complex, and vitamin C in the second trimester. As for the third trimester, the price for menu per day without supplements is US$1.65 and with supplements is US$1.59.

From the diet optimization model in this study, it was found that RNI for iron and vitamin B3 was hard to achieve without supplements or fortified foods. This finding is similar to reports from other studies [18,20]. This finding is further supported by the recommendation for iron consumption for pregnant women by MDG 2010 [14]. In developing countries such as Malaysia, iron deficiency has been identified as one of the main challenges for pregnant women [32]. Therefore, iron supplementation is prescribed to this population segment at no cost if the women are using public health facilities for their antenatal check-up. In this study, menus were suggested with and without iron supplementation for pregnant women to highlight the difference in costs for these two menus. The supplementation should reduce the daily food costs as iron has been identified as one of the nutrients that is quite challenging to be achieved, especially when the cost of foods is becoming one of the limiting factors for pregnant women in achieving healthy diet.

These supplements increased haemoglobin levels in pregnant women as seen in this study, where positive association was found between iron supplement consumption and haemoglobin level among pregnant women. Another study [25] also found that iron supplementation was associated with anaemia among pregnant women. In an article by Preziosi et al. [33] it is shown that iron supplement significantly reduced anaemia during pregnancy as well as the post-partum period. This is very important considering severe postpartum anaemia is a complication in at least 5% of deliveries. Even in normal deliveries, without any complications, following the deliveries, women lose some amount of iron during breastfeeding and lactation period. Iron deficiency anaemia was associated with impaired cognitive function and behavioural disturbances in postpartum women [33]. Iron deficiency persists beyond the 4–6 weeks postpartum period with 12% of women being iron-deficient up to 12 months after delivery and 8% of women being iron-deficient 13–24 months after delivery. Hence, iron supplementation should be continued after delivery if the iron status remains low or the mother is still breast feeding [33].

Ferrous iron salts (ferrous fumarate, ferrous sulphate and ferrous gluconate) are the preferred oral preparations of iron as it gives better bioavailability of elemental iron. Slow-release tablets are preferred as it is better tolerated, and absorption is 29% greater than the standard preparation. Iron supplements should be taken at bedtime or between meals to ensure optimum absorption [5]. WHO recommends 120 mg/day of elemental iron as a supplement for pregnant women [5]. The RNI 2017 recommendation for iron supplement is 100 mg/day of elemental iron for pregnant women [15]. Besides supplements provided by the Ministry of Health (MOH), pregnant women in Malaysia have many choices for iron supplements that can be bought from pharmacies and private clinics. However, the supplements used in this study for the diet models with supplements are the supplements provided by the MOH for free. Hence, the price of the menu per day would increase if the pregnant women buy supplements from pharmacies.

The strength of this study is the utilization of locally available foods in our diet optimization models to improve dietary intakes as the solution for key nutrient problems among pregnant women. While recommendations are generally useful, it seems that nutritional needs for pregnant women should be handled in the clinic or health centre on an individual basis based on overall needs and preferences of the patient. Individualized nutrition would be the best solution to ensure adequate dietary intake especially if the expecting mothers are experiencing health issues [34]. The final constructed menus contained a variety of ingredients such as wholegrain cereals (brown rice, maize flour, wheat flour, barley, oats); roots and tubers (potatoes); fish patin (silver catfish), sardines, selar (yellow tail scad), and anchovies; indigenous fruits (jackfruit, soursop); pulses, nuts and seeds; green and yellow vegetables; and meat that can be sourced locally. Also, the diet optimization models contained a mixture of ingredients that can enhance the bioavailability of iron. The fruits and vegetables, for example, citrus fruits and mustard spinach are rich in ascorbic acid—a well-known iron absorption enhancer. Fish and poultry meat can improve the bioavailability of iron and zinc in the constructed menus [35]. The seeds and nuts in the diet models can also provide essential oils that can enhance the bioavailability of beta-carotene in the menus [36]. On the other hand, the constructed menus may contain iron and nutrient absorption inhibitors such as phytates in wholegrain cereals, seeds and legumes. Experimental studies are required to establish the actual biological value of essential micronutrients in the diet optimization model. Any future changes in price and availability of local ingredients can be modified and accommodated using linear programming techniques used in the current study.

Several limitations of this study warrant mentioning. First, due to the limitation of the Malaysian food database, the current study did not estimate some nutrients relevant to maternal and foetal nutrition including fatty acids, folic acids, and vitamin D. Moreover, the optimized diets were built on the hypothesis that the observed diets reflect individual preference. In reality, many other factors may contribute to food choices. Thus, information about actual personal preference will clearly help to improve the approach. Future studies are needed to confirm the improvement of dietary quality through personalized nutrition recommendation as suggested by the linear programming approach, as compared to conventional, non-personalized dietary guidelines. Secondly, as this study was an exploratory study that aimed to investigate the possibility of developing a palatable and culture-specific diet plan, using the diet optimization model, which achieves the national recommendations with minimal cost for pregnant women in the selected area, convenience sampling made the results unlikely to be conclusive. Convenience sampling also limits the generalizability of the study findings across populations. Nevertheless, the study highlighted the importance of individualized nutrition particularly for pregnant women. Pregnant women, especially low-income, need to select food items wisely as limited food expenditure could become the main barrier to achieve healthy eating status. The lowest costs menus obtained from linear programming model could provide a direct comparison as whether or not healthy foods are affordable for the low income. Future studies should consider longitudinal study and random sampling so that causal relationships can be established as well as enhancing the generalizability of the findings. As this study only focused on one selected suburban location, future studies should also consider the rural and urban areas to represent the general population.

## 4. Conclusions

In conclusion, linear programming helped in translating nutrient recommendations into personal diet advices by providing diet optimization models that meet the Malaysian Dietary Guidelines and Recommended Nutrient Intake for a sample of suburban pregnant women from Selangor, Malaysia. Substantial changes including increasing diversity and intakes of nutrient-dense foods as well as iron and hematinic vitamins were needed to achieve optimal diets. Such information can prove useful when establishing dietary guidelines and meal plans for pregnant women, and when advocating a healthy diet with adequate nutrient supply for both foetuses and the mothers’ own nutrition storage.

## Figures and Tables

**Table 1 ijerph-16-04720-t001:** Sociodemographic characteristics and anthropometric data of pregnant women (*n* = 78).

Characteristics	*n*	%	Mean	Range
Age (years)				
20–29	31	39.9	31.24 ± 4.94	21–45
30–39	46	58.8		
40–49	1	1.3		
Ethnic				
Malay	77	98.7		
Indian	1	1.3		
Educational Status				
Secondary education	34	43.6		
Tertiary education	44	56.4		
Household Income				
<USD 241	4	5.1	2640 ± 879	900–5100
USD 241–USD 722	26	33.3		
USD 723–USD 1204	28	35.9		
>USD 1204	20	25.6		
Gravidity				
1	17	21.8	2.83 ± 1.507	1 to 7
2	19	24.4		
3	20	25.6		
4	10	12.8		
5	7	9		
6	4	5.1		
7	1	1.3		
BMI				
Underweight (<18.5 kg/M^2^)	10	12.8	24.3 ± 4.7	17.4–34.8
Normal (18.5–22.9 kg/M^2^)	32	41		
Overweight (23–27.4 kg/M^2^)	22	28.2		
Preobese (27.5–34.9 kg/M^2^)	14	18		
Hb Level				
Normal (11–16 g/dL)	55	70.5	11.1 ± 0.82	8.3–12.8
Mild anaemia (10–10.9 g/dL)	17	21.8		
Moderate anaemia (8–10 g/dL)	6	7.7		

**Table 2 ijerph-16-04720-t002:** Calories and nutrient intake with the percentage of subjects achieving the Recommended Nutrient Intake (RNI) (*n* = 78).

Nutrients	Mean Intake	Percentage of Subjects Achieving RNI (%)
Energy (Kcal)	2310 ± 292	48.7
CHO%	54% ± 5	96.1
Protein%	15% ± 2	87.1
Fat%	30% ± 4	39.7
Niacin (mg)	1.1 ± 0.2	11.5
Riboflavin (mg)	1.6 ± 0.4	41.0
Thiamin (mg)	20.2 ± 6.7	65.3
Vitamin C (mg)	125 ± 74	64.1
Vitamin A (RE)	1189 ± 630	80.8
Calcium (mg)	689 ± 154	2.6
Iron (mg)	23 ± 7	70.5
Phosphorus (mg)	1201 ± 234	97.4
Sodium (mg)	2813 ± 1011	98.7
Potassium (g)	2.5 ± 0.6	1.3
Fibre (g)	13 ± 4	5.1

**Table 3 ijerph-16-04720-t003:** Comparison of food groups by the models for each trimester with Malaysian Dietary Guidelines (MDG) 2010.

Constraint (Food Groups/Items)	LB* (Serving Size)	UB* (Serving Size)	Trimester 1	Trimester 2	Trimester 3
	MDG 2010		Diet Optimization Models		
Cereals and grains	4	8	6	6	7
Fruits	2	3	2	3	3
Vegetables	3	5	3	3	3
Meat/Poultry	0.5	2	1	2	1
Fish	1	3	1	1	2
Legumes	0.5	1	0.5	0.5	1
Milk and dairy products	1	3	2	2	2
Palatability					
Sugar	1	4	3	4	4
Salt	1	4	3	3	4
Oil	1	4	3	4	4

* LB—lower bound, UB—upper bound. One serving size—30 g carbohydrate for cereals, 5 g carbohydrate for sugar, 14 g protein for meat/poultry, fish, 7 g protein for legumes and milk, and 9 g fat for oil.

**Table 4 ijerph-16-04720-t004:** Nutrient content achievement of sample menu models according to trimester with the RNI 2017.

Nutrients	LB	UB	Trimester 1	LB	UB	Trimester 2	LB	UB	Trimester 3
Energy (kcal)	1680	1880	1806	1880	2080	2010.6	2070	2270	2215
Protein (g)	53	90.5	73.3	60.5	98	77.5	77.5	115	81.4
CHO (g)	185	305.5	241.1	210	338	275.6	233.8	368.9	303
Fat (g)	54	65	60.9	60	71	64.9	65	78	73.1
Fibre (g)	20	30	22.6	20	30	21.1	20	30	23.5
Vitamin A (RE)	800	2800	2195.6	800	2800	1353.1	800	2800	2102.5
Vitamin C (mg)	80	2000	270.3	80	2000	263.6	80	2000	270.3
Vitamin B1 (mg)	1.4	N/A	5.9	1.4	N/A	7.1	1.4	N/A	5.9
Vitamin B2 (mg)	1.4	N/A	3.9	1.4	N/A	6.9	1.4	N/A	3.9
Vitamin B3 (mg)	18	35	23.7	18	35	24.4	18	35	23.7
Sodium (mg)	1500	2300	2120.4	1500	2300	2105.8	1500	2300	2112.3
Potassium (mg)	4700	N/A	4703.5	4700	N/A	4862	4700	N/A	4882.2
Calcium (mg)	1000	2500	1144.7	1000	2500	1228.5	1000	2500	1458.2
Iron (mg)	20	29	23.1	20	29	24.2	20	29	28.5
Phosphorus (mg)	700	3500	1522.1	700	3500	1641.8	700	3500	1908.1

**Table 5 ijerph-16-04720-t005:** Development of a three-day menu without supplements and a one-day menu with supplements for pregnant women in the first trimester.

Meals	Day 1	Day 2	Day 3	With Supplements
Breakfast	Fried banana balls 6 pieces Coco drink 1 cup	Toasted bread + butter spread 2 slices Tea 1 cup with sugar	*Kampung* fried rice 1 plate (egg, beef, water spinach)	Kellogg’s froot loop breakfast cereal Full cream milk 1 glass
Morning Tea		Full cream milk 1 glass	Full cream milk 1 glass	Mung bean porridge
Lunch	Brown rice 2 scoops Chicken liver sambal Fried egg Wild pepper leaf Red-fleshed banana 1 whole	Brown rice 2 scoops Red spinach cooked in coconut milk Sambal sardine + potato Honey dew 1 slice	Brown rice 2 scoops Fried scad in chilli 1 piece Coconut stew, fern shoot Orange 1 whole Soybean milk 1 glass	Brown rice 2 scoops Silver catfish curry Fried mustard greens with mushroom Red-fleshed banana 1
Afternoon Tea		Mung bean cakes 2 pieces	Full cream milk 1 glass	Banana fritters 4 piece coco drink 1 cup
Dinner	Brown rice 2 scoops Silver catfish cooked with chilli Mushroom soup Fried Chinese kale	Fried rice noodles (mee hoon goreng) (chicken, chilli, sawi, tofu) Honey dew 1 slice	Beef burger 1 (beef patty 1, cabbage, bun) French fries Orange 1 whole Full cream milk 1 glass	Brown rice 2 scoops Chicken sambalPiper fried with egg
Supplements				Folic acid tablet
Calories	1841.8	1853	1813	1715.7
Food Cost Per Day (US$)	1.58	2.04	2.92	1.56

**Table 6 ijerph-16-04720-t006:** Development of a three-day menu without supplements and a one-day menu with supplements for pregnant women in the second trimester.

Meals	Day 1	Day 2	Day 3	With Supplements
Breakfast	Roti canai telur 1 slice Dhal gravy	Egg sandwich 2 slices Coffee 1 cup	Coconut milk rice (sambal udang, egg, cucumber, fried peanuts, fried achovies)	Friend onion balls (onion) 6 pieces Coco drink
Morning Tea	Full cream milk 1 glass	Barley drink 1 glass	Full cream milk 1 glass	Raisins 1/4 cup Full cream milk 1 glass
Lunch	Brown rice 2 scoops Silver catfish with chilli coconut stew Wild pepper leaf Lady finger banana 2 whole	Brown rice 2 scoops Pumpkin cooked in coconut milk Sardines fried with chilli Honey dew 1 slice	Chicken rice + roasted chicken Coconut water 1 glass Mango 1 whole	Brown rice 2 scoops Grilled Silver catfish Wild pepper leaf
Afternoon Tea	Raisins 1/2 cup	Full cream milk 1 glass	Full cream milk 1 glass	mung beans cake Barley drink 1 glass
Dinner	Brown rice 2 scoops Chicken liver cooked in soy sauce Fried mustard greens Lady finger banana 2 whole	Curry Noodles (chicken, tofu, bean sprout, prawn) Honey dew 1 slice Full cream milk 1 glass	Rice, cooked Grilled catfish Fried Chinese kale Orange 1 whole	Brown rice 2 scoops Egg cooked in chilli Mustard green soup Lady finger banana 2 whole
Supplements				Folic acid, iron B complex and vitamin C tablets
Calories	2105.6	1998.8	1954	1984.1
Food Cost Per Day (US$)	1.61	2.30	2.80	1.47

**Table 7 ijerph-16-04720-t007:** Development of a three-day menu without supplements and a one-day menu with supplements for pregnant women in the third trimester.

Meals	Day 1	Day 2	Day 3	With Supplements
Breakfast	Roti canai 2 pieces Dhal gravy	Pancake, wheat Tea 1 cupOrange 1 whole	*Kampung* fried rice Watermelon 1 slice	Roti jala 4 pieces Chicken curryRed-fleshed banana 1 whole
Morning	Jackfruit (nangka) 3 pieces Coco drink 1 cup	Peanuts fried with flour Full cream milk 1 cup	Full cream milk 1 glass	Raisins ¼ cup Full cream milk 1 glass
Lunch	Brown rice 2 scoops Silver catfish cooked with tamarind Wild pepper leaf	Brown rice 2 scoops Red spinach soup fried sardine Beansprout salad. Orange 1 whole	Brown rice Chicken kurma Ulam pegaga Soursop ½ cup	Brown rice 2 scoops Silver catfish with Wild pepper leaf Red-fleshed banana 1 whole
Afternoon Tea	Raisin bread 2 slices Full cream milk 1 glass	White bread + egg jam 2 slices Full cream milk 1 glass	Stuffed Beancurd 2 pieces Full cream milk 1 glass	Mung beans porridge, 1/2 bowl Coco drink
Dinner	Fried rice (mustard green, chicken, egg) Jackfruit 2 pieces	Fried wet noodles (sawi, chicken, shellfish) Honey dew 1 slice Full cream milk 1 glass	Rice, cooked Tilapia cooked in chilli Fried kalian Pineapple ½ cup Full cream milk 1 glass	Brown rice 2 scoops Chicken tomyam (+mushroom) Fried mustard green Ommelette 1
Supplements				Folic acid, iron, B complex, and vitamin C tablets
Calories	2208	2263.3	2209.1	2181
Food Cost Per Day (US$)	1.65	2.59	3.19	1.59

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
