# Peer review of "Utilization of a Diet Optimization Model in Ensuring Adequate Intake among Pregnant Women in Selangor, Malaysia"

_ijerph, 2019, doi:10.3390/ijerph16234720_

Round 1

Author Response

Thank you for your valuable feedback, comments and suggestion to our manuscript. We are really grateful to you for your insightful comments on our paper. We have been able to incorporate changes to reflect most of the suggestions provided. We have highlighted the changes within the manuscript. Attached are point-by-point response to the feedbacks, comments and suggestion. I have also send the manuscript to be proof read by an English Editor.

Author Response

Thank you for giving me the opportunity to submit a revised draft of my manuscript titled “The Utilization of Diet Optimization Model in Ensuring Adequate Intake Among Pregnant Women in Selangor, Malaysia”. We appreciate the time and effort that you have dedicated to providing your valuable feedback on our manuscript. We are grateful for the insightful comments on our paper. We have been able to incorporate changes to reflect most of the suggestions.

Here is a point-by-point response to the comments and concerns.

Reviewer 3 Report

In their manuscript, Hamid et. al. desrice their study to define a culturally specific, regional, palatable menu for Malay pregnant mothers which reaches the recommended dietary guidelines. The study is well written and the results clear; however, some consideration needs to be given to minor issues, and the authors need to spend a significant amount of time accounting for other sources of bias present in this type of cross-sectional study. Indeed, further data to reduce this bias would greatly improve the novelty and importance of this manuscript.

As mentioned very late in the manuscript (lines 272-274). While recommendations are generally useful, it seems that nutritional needs for pregnant women should be handled in the clinic on an individual basis based on overall needs and preferences of the patient. Can the authors further comment on this point? It should perhaps also be mentioned in the text that individualized nutrition would be the best solution to ensure adequate dietary intake. Much more consideration needs to be given to the bias present in this study. Indeed, for example, compared to the general Malaysian population described previously (https://www.ncbi.nlm.nih.gov/pmc/articles/PMC4438653/) -- this study population seems to have some bias in regards to BMI, especially versus overweight/obese and underweight. This can significantly change the results of the study as food intake is likely largely overexaggerated in the overweight/obese mothers versus those underweight. In addition, the choice to take a convenience sample at a suburban clinic indicates selection bias and further data from true urban and rural populations need to be considered before making statements about adequacy of diet for pregnant women in Malaysia. Please provide a power calculation used to ensure adequate input data for the study as n = 78 seems rather low to generate generalized nutritional guidelines. Abbreviations in tables need to be defined. Table 3 is extremely difficult to interpret and could use some revision. It is hard to understand what is being compared and the comparison result outcomes. Also, the RNI 2017 needs to be included if it is being used in the study. Lines 177-180: Please further define how semi-quantitative score for “palatability” was generated as it seems to be subject to major subjectivity across included subjects. How does the diet generated match to the palatability and food access to each participant?

Author Response

Thank you for your valuable feedback, comments and suggestion to our manuscript. We are really grateful to you for your insightful comments on our paper. We have been able to incorporate changes to reflect most of the suggestions provided. We have highlighted the changes within the manuscript.

Attached are point-by-point response to the feedbacks, comments and suggestion.

Reviewer 4 Report

I have now thoroughly read your paper entitled "The utilization of diet optimization model in ensuring sufficient dietary intake among pregnant mothers". I find the topic to be very interesting and the study to contribute to more knowledge regarding nutrient deficiency among pregnant women. However, I have some comments and suggestions (see below) that I think should be considered before I can recommend the paper for publication. My recommendation is thus "Major revision"

Comments and suggestions

Title:

Specify that this study deals with Malaysia. Suggestion: Add “in Malaysia” at the end of the title.

Abstract:

Page 1, line 22: “(list here)” – should something be inserted here? If not, remove

Introduction:

Page 1, line 40-41: In my opinion, the sentence “These morbidities are directly associated with unhealthy diet” is quite harsh and should be less definite. In particular, iron deficiency usually has multiple causes. Maybe change “unhealthy” with “nutritionally incomplete” or “inadequate”?

Page 2, line 58: Change “is” to “was” in the sentence “Hence, tha aim of this study is to provide…”

Materials and methods:

Page 2, line 64: Remove comma between the words “in” and “Selangor”

Page 2, line 66: “… less than or equals to parity of 6” – unclear meaning, please rephrase

Page 2, line 67: Insert “as” between the words “such” and “GDM”

Page 2, line 75: “…two weekdays and one weekend in a week” – unclear meaning, please rephrase

Page 2, line 85: “wet market” – what is this, please explain

Page 3, line 100: “RM” – since this article is meant for international readers, could you please relate this currency to an internationally more known currency like USD?

Page 3, line 106: “Besides, the upper limit and lower limit on limit portion size…” – unclear, please rephrase

Page 3, line 108-109: change “are” to “were” in “The food groups are grouped….”

Results and discussion:

Page 3, line 118: It says here (and also in table 1) that the participants’ age ranged between 21 and 45. On page 2, line 65 it says that the range was 20-45. Which is correct?

Page 3, line 119: Insert “The” before “Majority of the participants…”

Page 3, line 121: Change “have” with “had” in “…. remaining 25.6% have household income….”

Page 3, line 122: Insert “The” before “Majority of the participants…”

Page 4, line 131-133: Sentence starting with “Ninety-four percentage…” – meaning is unclear, please rephrase. Also in this sentence, change “percentage” with “per cent”.

Page 4, line 136: Here, you state that supplementation was statistically associated with anaemia – please state whether this association was positive or negative?

Page 4, line 144-145: Insert the word “weight” between “maternal” and “gain”

Page 4, line 147: Insert “the” between “while” and “majority”

Page 5, line 159: “26” should be in brackets [26]?

Page 5, line 172: Please define “serving size” – is this equal to number of portions? In that case, please define how many grams equals a portion?

Page 7, line 194-213: Please rephrase this section. In my opinion, the option of receiving nutritional supplement at antenatal controls should be mentioned earlier in the manuscript, maybe already in the introduction section. This because it is very common to take these supplements and will thus act as a pre-condition for many of the results and discussions in the paper.

Page 7, line 217: The reference of “Paul Preziosi” should not be referred like this. Please rephrase (suggested “In an article by Preziosi et al. it is showed….”)

Page 7, line 220-221: Please rephrase. It may be that mothers lose iron or have iron deficiency in the breast-feeding period, but it is not because the iron is “leached” into the milk, as the iron content of human milk is very low.

Page 7, line 223-224: change the word “are” to “being” in the sentence “…12% of women are iron deficient…” and “… 8% of women are iron deficient…”

Page 10, line 270: Remove the word “may” in the sentence “… the observed diets may reflect individual…”

Tables:

Table 1:

Under “Gravidity” – the number 1 is referred twice, please remove

Are BMI and Hb levels during pregnancy/antenatal controls or from before the pregnancy? Please clarify

Under “Hb Level” – Change “Moederate” to “Moderate”

Table 3:

Please define “serving size”

Table 4:

Please change to “landscape” orientation instead of “portrait” – it will be easier to read

Reference list:

References where organizations are authors, make sure that “author name” is presented right (e.g. ref 7 and 19)

Author Response

We appreciate the time and effort that you have dedicated to providing your valuable feedback on our manuscript. We are grateful for the insightful comments on our paper. We have been able to incorporate changes to reflect most of the suggestions.

Here is a point-by-point response to the comments and concerns.

Round 2

Reviewer 3 Report

The authors have adequately addressed my previous concerns. Thank you.

Author Response

Dear Sir/Madam,

I have sent the article to an English proof reader and she have checked and corrected the grammar and the spellings of the articles.

Thank you

Reviewer 4 Report

I have now thoroughly read your manuscript entitled "The utilization of diet optimization model in ensuring adequate intake amoung pregnant women in Selangor, Malaysia". I think the manuscript is of good quality. However, I have attached some comments and suggestions below that I think will make it a bit clearer. My recommendation is thus "Accept after minor revision"

Comments and suggestions

Page 1, line 33: “Shows” instead of “showed”

Page 1, line 35: “Women” instead of “mothers”

Page 2, line 66: “Women” instead of “mother”

Page 2, line 66: Remove “had” in the sentence “… period and had parity…”

Page 2, line 75: Rephrase to “… three days in a week including either Saturday or Sunday”

Page 2, line 83-84: Remove the abbreviations RNI 2017 and MDG 2010, as you have already explained these.

Page 3, line 97: “that” instead of “which”

Page 3, line 122: “with” instead of “and received”

Page 4, line 135: “these” instead of “this number”

Page 4, line 140: Remove “of them” in the sentence “… anaemic group, 79 % of them did not consume…”

Page 4, line 141-143: I think this should be rephrased, as it is now it sounds like iron supplementation leads to anaemia… I would write that “… pregnancy was positively associated with Hb level (p < 0.05).” Also, in the next sentence, “… statistically associated with Hb level among pregnant women.”

Page 4, line 142: Remove “vitamin” – iron is not a vitamin

Page 4, line 148: “Zahara” is the author’s first name, please change to “Manaf”

Page 5, line 154: “Yue” is the author’s first name, please change to “Cheng”

Page 5, line 166: “Zahara” is the authors first name, please change to “Manaf”

Page 5, line 170: “with” instead of “which is”

Page 5, line 175: “fulfilment” instead of “satisfaction”

Page 5, line 181: Remove the parenthesis at the end of the sentence

Page 7, line 222, 223, 225: “women” instead of “mothers”

Page 7, line 248: MOH – please write the full name (first time mentioned)

Page 10, line 294: “guidelines” instead of “guideline”

Page 10, line 294-297: Very long sentence that is difficult to understand. Please rephrase.

Page 10, line 300: “women” instead of “mothers”

Author Response

Dear Sir/Madam

We are very grateful for the comments and suggestions  provided by you for this manuscript. I have sent the manuscript to an English Proof Reader to improved the english language and spelling. Please see attachment, in red, our detailed response to comments.
